# Glycerol Trinitrate Acts Downstream of Calcitonin Gene-Related Peptide in Trigeminal Nociception—Evidence from Rodent Experiments with Anti-CGRP Antibody Fremanezumab

**DOI:** 10.3390/cells13070572

**Published:** 2024-03-25

**Authors:** Nicola Benedicter, Birgit Vogler, Annette Kuhn, Jana Schramm, Kimberly D. Mackenzie, Jennifer Stratton, Mária Dux, Karl Messlinger

**Affiliations:** 1Institute of Physiology and Pathophysiology, Friedrich-Alexander-University, D-91054 Erlangen, Germany; nicolabenedicter@gmail.com (N.B.);; 2Teva Pharmaceuticals, Redwood City, CA 94063, USA; kimberly.mackenzie01@gmail.com (K.D.M.); jennifer_stratton@mac.com (J.S.); 3Department of Physiology, University of Szeged, H-6720 Szeged, Hungary; dux.maria@med.u-szeged.hu

**Keywords:** fremanezumab, monoclonal antibody, calcitonin gene-related peptide, glycerol trinitrate, CGRP release, CGRP concentration, rat, migraine pain

## Abstract

Calcitonin gene-related peptide (CGRP) and nitric oxide (NO) have been recognized as important mediators in migraine but their mechanisms of action and interaction have not been fully elucidated. Monoclonal anti-CGRP antibodies like fremanezumab are successful preventives of frequent migraine and can be used to study CGRP actions in preclinical experiments. Fremanezumab (30 mg/kg) or an isotype control monoclonal antibody was subcutaneously injected to Wistar rats of both sexes. One to several days later, glyceroltrinitrate (GTN, 5 mg/kg) mimicking nitric oxide (NO) was intraperitoneally injected, either once or for three consecutive days. The trigeminal ganglia were removed to determine the concentration of CGRP using an enzyme-linked immunosorbent assay (ELISA). In one series of experiments, the animals were trained to reach an attractive sugar solution, the access to which could be limited by mechanical or thermal barriers. Using a semi-automated registration system, the frequency of approaches to the source, the residence time at the source, and the consumed solution were registered. The results were compared with previous data of rats not treated with GTN. The CGRP concentration in the trigeminal ganglia was generally higher in male rats and tended to be increased in animals treated once with GTN, whereas the CGRP concentration decreased after repetitive GTN treatment. No significant difference in CGRP concentration was observed between animals having received fremanezumab or the control antibody. Animals treated with GTN generally spent less time at the source and consumed less sugar solution. Without barriers, there was no significant difference between animals having received fremanezumab or the control antibody. Under mechanical barrier conditions, all behavioral parameters tended to be reduced but animals that had received fremanezumab tended to be more active, partly compensating for the depressive effect of GTN. In conclusion, GTN treatment seems to increase the production of CGRP in the trigeminal ganglion independently of the antibodies applied, but repetitive GTN administration may deplete CGRP stores. GTN treatment generally tends to suppress the animals’ activity and increase facial sensitivity, which is partly compensated by fremanezumab through reduced CGRP signaling. If CGRP and NO signaling share the same pathway in sensitizing trigeminal afferents, GTN and NO may act downstream of CGRP to increase facial sensitivity.

## 1. Introduction

The neuropeptide calcitonin gene-related peptide (CGRP) and nitrogen species such as nitric oxide (NO) are potent vasodilatory agents and important mediators in migraine [1,2]. During spontaneous migraine attacks, elevated concentrations of CGRP have been measured in venous plasma [3,4,5], saliva [6,7], and tear fluid [8]. Inhibiting CGRP release with triptans and ditans; targeting CGRP using monoclonal antibodies (mAbs) against the CGRP ligand; or blocking CGRP receptors by the CGRP receptor antibody or gepants are effective treatments in migraine [9,10]. Notably, the infusion of CGRP can provoke short-lasting headaches, which are attributed to the vasodilatation of cranial blood vessels and, additionally, delayed migraine-like pulsating pain in migraineurs [11,12]. 

Similarly, elevated plasma concentrations of nitrogen metabolites have been found in plasma during migraine [13], and the infusion of nitroglycerin (glycerol trinitrate, GTN), which mimics NO, has been shown to cause transient vasodilatory headaches and, in addition, migraine-like pain in migraineurs [14,15]. The inhibition of endogenous NO production and NO-mediated mechanisms have not been used for migraine therapy, partly due to adverse side effects [16]. Due to the similarities of CGRP and GTN administration in headache provocation, there might be an association between endogenous CGRP and NO actions. The question is whether and how these two mediator systems may be linked. GTN-induced migraine-like attacks have been found to be associated with an increase in plasma CGRP levels [17], which responds to treatment with sumatriptan [18,19], suggesting CGRP release provoked by NO mechanisms. However, an increase in plasma CGRP during GTN-induced headache could not be verified by another group [20]; the headache was not prevented by blocking CGRP receptors [21]. Thus, a causal link between CGRP and NO signaling in migraine is still an open question.

To generate models of migraine related to the mentioned clinical experiments, several groups have administered NO donors or GTN in rodents and collected multiple structural, functional, and behavioral data [22,23,24,25,26]. Our group has recently examined the effects of fremanezumab, an anti-CGRP monoclonal antibody (mAb) used for the prevention of chronic and frequent migraine, on meningeal CGRP release and blood flow, CGRP concentration in trigeminal ganglia, and facial sensitivity [27,28,29]. To compare these data in an animal model of trigeminal nociception, we have repeated some of these experiments after the administration of GTN and found deviating results, which may contribute to the question of how endogenous CGRP and NO signaling may be linked. Our data provide evidence that CGRP signaling is bypassed by NO signaling, which favors the hypothesis that NO may act downstream of CGRP signaling in migraine pathophysiology.

## 2. Materials and Methods

Animal housing and all experiments were carried out according to the regulations for the care and treatment of laboratory animals of the European Communities Council Directive 1986 (86/609/EEC), amended 2010 (2010/63/EU). The experimental protocols were reviewed by an ethics committee and approved by the District Government of Middle Franconia (54-2532.1-21/12).

### 2.1. Animals

Fifty-two adult Wistar rats of both sexes (body weight of females: 220–370 g; males: 310–450 g), bred and housed in our animal facility, were used. Groups of 3–4 animals were kept in cages in a 12 h day–night cycle and received pellet food and water ad libitum. Equal numbers of animals were matched according to their sex and weight and allocated to the treatments. The estrus state of the females was not assessed. 

### 2.2. Administration of Monoclonal Antibodies

Administration of mAbs was performed as previously reported [27,28,29]. In short, either 30 mg/kg of the anti-CGRP mAb fremanezumab, or the isotype control mAb (both Teva Pharmaceuticals, Redwood City, CA, USA) diluted in saline (10 mg/mL), was subcutaneously injected into the neck of the animals under short isoflurane anesthesia. Fremanezumab binds to an epitope that is identical in human α-CGRP, β-CGRP, and rat β-CGRP. There is one similar amino acid change within the epitope for rat α-CGRP, which has a minor effect on overall binding affinity. The operator was blinded to the identity of the antibodies. The animals were individually marked, placed back in their cages, and inspected daily regarding their health state and behavior until they were used for further procedures.

### 2.3. Administration of GTN and Dissection

On day 1, 3, 10, or 30 after the antibody injection the animals received an intraperitoneal (i.p.) injection of 5 mg/kg GTN (Nitrolingual, containing 1 mg/mL GTN, Pohl-Boskamp, Hohenlockstadt, Germany) or the same volume of synthetic interstitial fluid (SIF) as the vehicle [27] under short isoflurane anesthesia at 8 a.m. (Figure 1A). Four hours later, the animals were deeply anesthetized and sacrificed in a rising CO_2_ atmosphere. Another group of animals, which were used for behavioral testing as outlined below, received, on three consecutive days, 5 mg/kg, 2.5 mg/kg and 1.25 mg/kg GTN, namely on days 4–6 and 11–13 after mAb administration, for a total of 6 applications (Figure 1B). With this design, a cumulative sensitizing effect of GTN causing chronic hyperalgesia [30] should be avoided. Animals were sacrificed on day 14 after mAb administration, 10 days after the first, and one day after the last GTN injection. The heads of the animals were separated, skinned, and cut into halves along the sagittal line, as previously described for CGRP release measurements [27]. The trigeminal ganglia were excised from the skull base at a length of 6–7 mm, placed in Eppendorf cups, and frozen at −20 °C until further processing. From each animal, both trigeminal ganglia were harvested, resulting in 104 samples.

### 2.4. Preparation and CGRP Determination

The processing of tissue samples and CGRP measurements were performed as previously reported [29]. In short, after defrosting, the trigeminal ganglia were dipped in an absorbent tissue, weighed, and immersed in 1 mL of 2 M acetic acid, boiled at 95 °C for 10 min, and homogenized. The homogenate was again boiled for 10 min and centrifuged at 2000 rpm. The supernatant was collected, and 100 µL thereof was diluted 4:1 with enzyme-linked immunosorbent assay (ELISA) buffer and processed with an ELISA kit for CGRP with a detection level of 2 pg/mL, according to the instructions of the manufacturer (Bertin Pharma/SPIbio, Montigny le Bretonneux, France). The CGRP concentration (in ng/mg) of each trigeminal ganglion was calculated with regard to the respective ganglion mass.

### 2.5. Preparation of Animals for Behavioral Tests

The preparation and performance of behavioral tests have previously been reported in detail [28] and are outlined here in short. After priming the animals for an attractive 10% sucrose solution available in standard drinking bottles, the animals were placed in a test cage in which they had to pass with their forehead through an opening; it could be equipped with a mechanical barrier (flexible 0.09 mm steel bristles) or a thermal barrier (circular tube with 50 °C heated water). The opening of the drinking bottle with the sucrose solution could be reached by the animal while its cheeks and forehead touched the steel bristles or the hot tube. The frequency and duration of passing through the opening were recorded automatically with a photo sensor device (Orofacial Stimulation Test System, Ugo Basile, Lugano, Italy) within test periods of 15 min. The volume of consumed sucrose solution during the test periods was determined afterward. 

### 2.6. Sequence of Testing

The test sequence started after 7 days of priming and was always conducted between 12 a.m. and 1 a.m. On the first test day (day-3), the animals could reach the sugar source without a barrier; on the second day (day-2), they had to pass through the mechanical barrier; and on the third day (day-1), they had to pass through the thermal barrier during the 15 min test period (Figure 1B). This provided a baseline measure for the 3 conditions before the animals received fremanezumab or the control mAb on the following day. The experimenter was blinded as to the specific mAb treatment during the whole experiment. On days 4–6 after mAb administration, the animals were injected with GTN as described, followed by the same test sequence 4 h later (Figure 1B). The whole test sequence, including the injection of GTN, was repeated, beginning with day 11 after mAb injection. 

### 2.7. Data Processing and Statistics

Statistical analysis was performed using Statistica software (StatSoft, Release 7, Tulsa, OK, USA). After verification of the normal distribution of data, analysis of variance (factorial ANOVA) with the categorical predictors (factors) mAb, sex, and day after mAb administration, as well as treatment (GTN or saline), was used for a comparison of data. This was extended by Tukey’s honest significant difference (HSD) test or the unequal N HSD test. In the case of the behavioral experiments, where the 3 barrier conditions are extremely different situations, data were compared for each condition separately using repeated measures ANOVA for the 3 days of the same tests, with sex and mAb as factors. Sample size calculation was based on the biometric planning for previous measurements [27,28] in accordance with the ethics approval mentioned above. The level of significance was set at *p* < 0.05. Data were displayed as mean ± SEM (standard error of the mean).

## 3. Results

### 3.1. CGRP Content of Trigeminal Ganglia 

The experiments included 26 female and 26 male rats, either treated with fremanezumab or with the isotype control mAb (13 females and 13 males in each group). The injection of GTN and the final experiment were performed after waiting 1 day in 9 animals; 3 days in 8 animals; 10 days in 13 animals; and 30 days in 10 animals following mAb administration. In further experiments, 12 animals (10 of them also used for behavioral experiments; see below) were treated with GTN on days 4–6 and again on days 11–13 after mAb administration. 

#### 3.1.1. General Observations

After mAb administration of either type, none of the animals showed unusual behavior or any other sign of disturbance. Both groups continuously gained weight dependent on the waiting time but without difference between animals treated with fremanezumab (*n* = 26; final body weight 333.6 ± 9.1 g) or the control antibody (*n* = 26; 333.1 ± 7.9 g). As expected, the body weight was significantly different between the sexes (factorial ANOVA, F_1,84_ = 194.2, *p* < 0.0001). During the hours after injection of GTN, in particular after repetitive administration, the animals appeared generally less active but showed no signs of spontaneous pain. 

#### 3.1.2. CGRP Concentration in Trigeminal Ganglia

In the 104 samples of trigeminal ganglia, factorial ANOVA with the factors mAb (fremanezumab vs. isotype control mAb), sex (female vs. male animals), and time after mAb injection (1, 3, 10, 30 days, and day 14 after repetitive GTN injection) was performed. Surprisingly, ANOVA showed slightly higher ganglion weights in females compared to males (F_1,84_ = 5.0, *p* < 0.05) and clearly significant differences between the days after mAb administration (F_4,84_ = 13.9, *p* < 0.0001). Calcitonin gene-related peptide concentration was calculated using the measured CGRP content and the ganglion mass, as noted in the Materials and Methods section. The CGRP concentration was significantly different between the sexes (factorial ANOVA, F_1,84_ = 38.8, *p* < 0.0001) and between the days after mAb administration (F_4,84_ = 27.9, *p* < 0.0001) but not between the two mAbs (F_1,84_ = 0.1, *p* = 0.74). The Tukey post hoc test indicated that the difference between days was mainly due to the unusually low ganglion mass of day-30 animals (in both the fremanezumab and the control mAb group) and the significantly lower CGRP concentration in ganglia treated repetitively with GTN. Detailed data are displayed in Table 1. 

#### 3.1.3. Comparison of CGRP Concentration with Previous Data from Animals Not Treated with GTN

The CGRP concentration of the trigeminal ganglia from the 30 animals sacrificed 1, 3 or 10 days after administration of the mAbs was compared with previous data from 26 rats with a similar distribution of sexes and mAbs [29]. These animals were not injected with GTN, and the 52 trigeminal ganglia were prepared in the same way as described for the present experiments. Data from these previous and present experiments were compared using factorial ANOVA with the factors sex, mAb type, time after mAb injection (1, 3, or 10 days), and treatment (GTN vs. no GTN). The ganglion masses were significantly different between the sexes (F_1,88_ = 5.3; *p* < 0.05) but not significant between the mAbs (F_1,88_ = 2.9; *p* = 0.09) and the days after mAb administration (F_2,88_ = 3.1; *p* = 0.05) or between the treatments (F_1,88_ = 0.1; *p* = 0.70). The CGRP concentrations tested with factorial ANOVA were clearly different between the sexes (F_1,88_ = 50.9; *p* < 0.0001) but not between the mAbs (F_1,88_ = 0.16; *p* = 0.69). The difference between the days was significant (F_2,88_ = 3.2; *p* < 0.05), while there was a clear difference between the treatments (F_1,88_ = 18.9; *p* < 0.0001), which was mainly due to the male fremanezumab animals (Figure 2).

### 3.2. Behavioral Experiments

Twenty animals (10 females and 10 males) were included in the study. One day after the baseline tests on three consecutive days, namely without a barrier, with a mechanical barrier, and with a thermal barrier, the animals received either fremanezumab or the isotype control mAb (see Figure 1B). On days 4–6 and again on days 11–13 after the mAb administration, the animals were treated with GTN as outlined in the methods and tested again in the same sequence. Data are displayed in Table 2 in detail.

#### 3.2.1. Number of Approaches to Source 

The number of approaches (counts) to the attractive sugar solution within 15 min was not significantly different between baseline (before mAbs administration) and on days 4–6 and days 11–13 after mAbs administration, i.e., neither without barrier nor with mechanical or thermal barrier. There was also no difference between the two mAbs and sexes.

#### 3.2.2. Time Staying at the Source 

The cumulative time spent at the source to consume the sugar solution was not significantly different on the three days under the three barrier conditions. However, solely under the mechanical barrier condition, repeated measures ANOVA indicated a weak difference between the two mAbs (F_1,32_ = 7.6, *p* < 0.05) and the sexes (F_1,32_ = 5.0, *p* < 0.05), which could not be attributed to a specific group by the post hoc HSD test. 

#### 3.2.3. Consumed Volume 

The consumed volume of sugar solution was again not significantly different at the three days under the three barrier conditions. Repeated measures ANOVA indicated a difference between the mAbs (F_1,32_ = 7.2, *p* < 0.05) under the mechanical barrier, which was not due to a specific group.

#### 3.2.4. Comparison of Behavioral Data with Animals Not Treated with GTN: No Barrier Condition

The behavioral measurements were compared with previous data from 12 rats with an equal distribution of sex and mAbs [28]. Without a barrier, animals treated with GTN tended to approach the attractive source less frequently, stayed for less time at the source, and consumed less sugar solution than animals not treated with GTN (Figure 3A–C). Repeated measures ANOVA with the factors day (baseline vs. days 4 and 11 after mAb injection), treatment (no GTN vs. GTN), sex (male vs. female animals), and mAb (control mAb vs. fremanezumab) indicated no difference between the three days regarding the number of approaches (F_2,72_ = 1.7, *p* = 0.19), but the time at the source (F_2,72_ = 7.8, *p* < 0.001) and the consumed solution (F_2,72_ = 16.1, *p* < 0.0001) increased significantly. This can easily be explained by an increase in body weight of about 45 g on average during the two weeks between the baseline measurements and the second test sequence (see Figure 1B). In addition, the difference between the treatments (GTN vs. no GTN) was highly significant regarding both time at the source (F_1,36_ = 10.6, *p* < 0.005) and the consumed volume (F_1,36_ = 24.1, *p* < 0.0001), but there were no significant differences between sexes and mAbs regarding any of these measurements.

#### 3.2.5. Comparison of Behavioral Data with Animals Not Treated with GTN: Barrier Condition 

With a mechanical barrier, animals treated with GTN tended to approach the attractive source less frequently when they received the control mAb (Figure 4A) but more frequently when they received fremanezumab (Figure 4A–C). GTN animals also tended to stay less time at the source and to consume less sugar solution than animals not treated with GTN (Figure 4B,C). Repeated measures ANOVA with the factors day (baseline vs. days 4 and 11 after mAb injection), treatment (no GTN vs. GTN), sex (male vs. female animals), and mAb (fremanezumab vs. control mAb) indicated differences between the three days regarding the number of approaches (F_2,72_ = 3.3, *p* < 0.05), the time spent at the source (F_2,72_ = 9.8, *p* < 0.0005), and the consumed solution (F_2,72_ = 9.9, *p* < 0.0005). The increase in time at the source and the consumed volume can be explained by the increase in body weight during the two weeks between baseline and day 12. Between the sexes and the treatments, no significant difference appeared. Between the two mAbs, the time spent at the source (F_1,36_ = 7.6, *p* < 0.01) and the consumed volume (F_1,36_ = 7.1, *p* < 0.05) were significantly different.

## 4. Discussion

### 4.1. Impact of GTN on the CGRP Concentration of Trigeminal Ganglia 

In rodent models of migraine or facial pain, mostly 10 mg/kg GTN have been applied as a single dose [31,32], or 5 mg/kg has been administered repetitively every second day [30,33]. Repetitive GTN injection in rats has been reported to induce spinal hyperalgesia and orofacial allodynia along with an increase in CGRP expression [30]. Our results show that the administration of a single dose of 5 mg/kg GTN increased the production of immunologically detectable CGRP in the trigeminal ganglion within four hours. This appears to be a short time for neuropeptide expression and formation, but it is consistent with earlier experiments in our group, wherein anesthetized rats after an i.v. infusion of GTN at 250 µg/kg for two hours and a waiting time of four extra hours [34], or of GTN at 1 mg/kg for two hours and a waiting time of two extra hours [35], was followed by an increased number of CGRP immunoreactive neurons counted in the trigeminal ganglion. In contrast to these results, in our present experiment, a multiple-dose administration of GTN over some days decreased the CGRP concentration in the examined trigeminal ganglia. 

The opposite observation of significantly lower CGRP concentrations was made in the trigeminal ganglia of animals repetitively treated with GTN (see Figure 2C). This may have been due to a continuous depletion of CGRP during the six days of treatment. The assumption that GTN treatment causes slow continuous CGRP release from trigeminal afferents is substantiated by our previous finding that the basal (unstimulated) CGRP release from the dura mater was also lower in animals treated with GTN, possibly through draining of the CGRP stores of peptidergic afferents in the dura mater [27]. In contrast, the CGRP release evoked by capsaicin superfusion was higher, indicating a sensitizing effect of GTN on the stimulated CGRP release. However, in animals that received fremanezumab, there was no difference between the vehicle and GTN, pointing to a blocking effect of fremanezumab on the GTN-induced increase in stimulated CGRP release [27]. Different from this result, fremanezumab seems not to influence the proposed GTN-induced increase in CGRP concentration in the trigeminal ganglion, as is indicated by the statistical analysis of animals that received fremanezumab or the control mAb (see Figure 2A,B). 

### 4.2. Sex-Dependent CGRP Concentration of Trigeminal Ganglia 

The mean CGRP concentration of ganglia in GTN-treated rats was lower in female compared to male animals (see Table 1 and Figure 2A,B). This confirms our previous experiments in animals not treated with GTN, where both CGRP content and concentration were lower in females [29]. The reason for this sex difference is not clear. It is likely that sex hormones influence the expression or production of CGRP. It has long been reported that in rat dorsal root ganglia, the number of CGRP immunoreactive neurons was lower in females and increased in ovariectomized animals but decreased in estradiol-treated animals [36]. More recent experiments showed that 17β-estradiol decreased CGRP plasma levels and CGRP release from isolated trigeminal ganglia in male rats [37]. However, generally, the CGRP concentration tended to be higher in the trigeminal ganglia of animals treated with GTN compared to non-treated animals, which is prominent after 3 days of mAb administration (see Figure 2C). The reason for this effect could be shrinkage of the ganglion mass (e.g., by fluid loss) or an increase in CGRP expression after GTN treatment. Indeed, post hoc statistical analysis showed that there was a significant interaction between sex and treatment regarding the ganglion weight, which was lower in GTN-treated male, but not female, rats (factorial ANOVA with factors sex and treatment, F_1,108_ = 11.76, *p* < 0.001; Tukey’s post hoc test, *p* < 0.05 for males but *p* = 0.28 for females). Nevertheless, the results indicate that GTN treatment leads to an increase in the expression of, or at least immunologically detectable, CGRP in the trigeminal ganglion. 

### 4.3. No Impact of Fremanezumab on the CGRP Concentration of Trigeminal Ganglia 

We have seen no significant difference in CGRP concentration between the ganglia of rats treated with fremanezumab versus the control antibody, which is consistent with our previous findings, although it contrasts with a decrease in CGRP immunoreactive trigeminal ganglion neurons [29]. The lack of change in CGRP concentration is reminiscent of a recent clinical study in which plasma CGRP levels in patients were determined during the treatment with fremanezumab and four months after discontinuation of the treatment [38]. CGRP plasma concentrations did not differ between during treatment and post-treatment and were on the same level as in healthy controls. Thus, it seems likely that the main therapeutic effect of monoclonal antibodies directed against CGRP depends rather on a decrease in CGRP release during the strong stimulation of trigeminal afferents, as we have demonstrated earlier [27]. On the other hand, long-term effects of anti-CGRP monoclonal antibodies may be based on a decrease in CGRP receptors, which is suggested by a significant reduction in trigeminal ganglion neurons immunoreactive to the CGRP receptor components RAMP1 and CLR [29]. 

### 4.4. Limitations Regarding the Calculation of CGRP Concentrations

The CGRP concentration of the tested trigeminal ganglia depends on the amount of CGRP but also the size of the ganglia. We did not remove the dura mater covering each ganglion in order to avoid the destruction of neurons and loss of CGRP during preparation. Therefore, a confounding factor could be the differing amount of connective tissue adhering to the ganglion that influences the weight and hence the CGRP concentration. We nevertheless regard the concentration as a reliable measure for the amount of CGRP, because in our previous study, we saw that there is a high positive correlation between the CGRP content and concentration of each ganglion [29]. 

### 4.5. Impact of GTN on Rat Behavior 

The behavioral data, in particular the number of approaches to the sugar source, may substantially be influenced by the general activity of animals. The time spent at the source, and particularly the consumed sugar solution, likely depend on the size of the animals, which is reflected by the general increase in these parameters during the 15 days between baseline measurements and day 11/12 after mAb administration (see Figure 2B,C and Figure 3B,C). In addition, GTN treatment seems to have a major impact on these behavioral parameters. 

Without a barrier, animals treated with GTN tended to approach the attractive source less frequently, stayed less time at the source, and consumed less sugar solution than animals not treated with GTN (Figure 2A–C). This may be the result of a generally lower activity after GTN, as mentioned above. Alternatively, the GTN effect may be due to a decreased attractiveness of the sugar solution, which is reminiscent of the loss of appetite and nausea during migraine attacks. GTN-induced trigeminal nociception has been used as an animal model for migraine, at least for migraine-related symptoms [24,25], and loss of appetite or nausea is a well-known symptom of migraine. Anti-CGRP antibodies have been found to improve not only the headache severity but also the autonomic symptoms including loss of appetite during migraine attacks [39]. However, because there was no difference between animals that received fremanezumab or the control mAb in the measured parameters of our experiments, the activity factor seems more likely for the decrease in functions. 

With a mechanical barrier, like without a barrier, animals treated with GTN tended to stay less time at the source and to consume less sugar solution than animals not treated with GTN (Figure 3B,C), which may have similar reasons as discussed above. However, different to the condition without barrier, animals treated with GTN tended to approach the attractive source more frequently when they received fremanezumab, spent more time at the source, and consumed more sugar solution than animals that received the control mAb (Figure 3A–C). In our previous publication on rats not treated with GTN, we interpreted this observation as a reduced facial sensitivity under the action of the anti-CGRP mAb. Facial hypersensitivity has frequently been observed in animal studies following the administration of GTN [24,40,41]. Therefore, following treatment with GTN, the increase in all three parameters (approaches, time, and intake of solution) in animals that received fremanezumab compared to the control mAb may result from reduced facial hypersensitivity. Reduced incidences of cephalic hyperalgesia/allodynia have also been reported in patients treated with anti-CGRP mAbs like galcanezumab or fremanezumab [42,43]. 

### 4.6. Summary of GTN Effects 

Taken together, GTN seems to increase the amount of immunologically detectable CGRP in the trigeminal ganglion, but fremanezumab seems not to significantly interact with this GTN effect. Fremanezumab had also no significant effect on the measurable CGRP concentration in trigeminal ganglia without GTN treatment, although in previous experiments, the number of CGRP containing trigeminal ganglion neurons, identified by immunofluorescence, was decreased in animals that received fremanezumab compared to the control mAb [29]. 

Regarding the behavioral data, GTN treatment generally tended to suppress the activity of the animals, with the final result being that they consumed less of the rewarding sugar solution. However, under the condition of an uncomfortable and irritating barrier, when the animals tried to reach the attractive source more often (but nevertheless consumed less of the solution), fremanezumab seemed to compensate partly for the depressing effect of GTN. The fremanezumab effect may result from its ability to decrease facial sensitivity so that the hypersensitivity caused by GTN is restricted. These results are in line with GTN treatment as an animal model for facial hyperalgesia/allodynia in migraine [24,40,41]. 

In addition, the results of this study show that the GTN effects are partly independent of manipulating the CGRP signaling system by anti-CGRP mAbs, although the extent of the GTN-induced hypersensitivity seems to be limited by reduced CGRP signaling. The latter is in accordance with previous studies in rodents showing that sumatriptan (lowering CGRP release) and olcegepant (blocking CGRP receptors) alleviate cranial and hind-paw hypersensitivity [44,45,46]. Performing an extended series of elegant experiments with cell cultures supplemented by measurements of facial sensitivity in mice, it has been recently reported that CGRP acts on glial cells (Schwann cells), which in turn produce NO species that are able to activate adjacent trigeminal afferent neurons [47]. The decisive NO species thereby may be nitroxyl (HNO), a redox sibling of NO formed in the presence of H_2_S, which is able to directly activate transient receptor potential channels of the ankyrin type (TRPA1) in trigeminal afferents [48,49]. NO species have recently also been suggested to activate transient potential receptor channels of the vanilloid type (TRPV1) through S-nitrosylation [50]. In any case, the classical intracellular NO-mediated mechanism increasing cyclic guanosine monophosphate [51] is not necessary to explain the activating and sensitizing effects of NO species in pain and migraine generation [52,53]. 

## 5. Conclusions

Considering the shared pathways in sensitizing the trigeminal afferent system between CGRP and GTN or NO species, we conclude that GTN and NO act downstream of CGRP to enhance facial sensitivity. The hypothesized NO-induced activation of sensory neurons, as proposed by the previously mentioned group [40], may complete a feedback loop if these neurons release CGRP. It is important to determine in human studies whether this hypothesis holds true for CGRP-NO signaling in migraine pathophysiology. This might explain why the CGRP receptor antagonist olcegepant has been reported not to prevent GTN-induced migraine [21]. 

## Figures and Tables

**Figure 1 cells-13-00572-f001:**
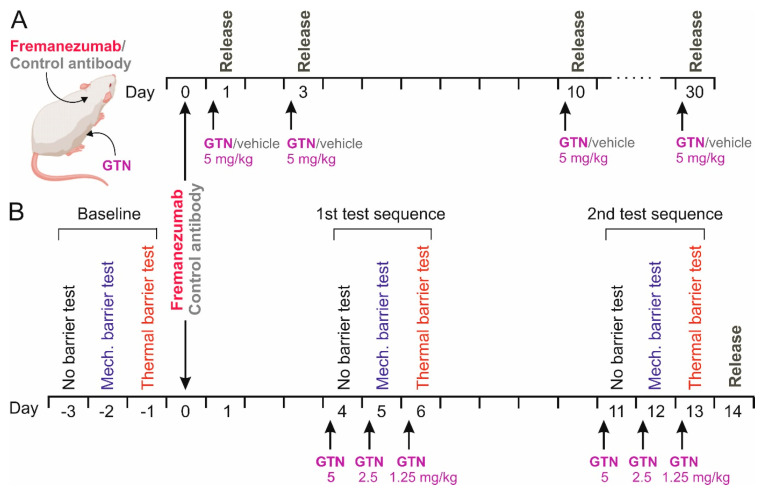
Schematic workflow of the CGRP release experiments (**A**) and the behavioral experiments followed by CGRP release (**B**). Either anti-CGRP mAb fremanezumab or control monoclonal antibody was subcutaneously injected, in B after baseline recordings without barrier, under mechanical and under thermal barrier conditions. Release experiments were performed 4 h after intraperitoneal injections of glycerol trinitrate (GTN) in **A** or after two test sequences according to the baseline recordings and repetitive injections of GTN in **B**.

**Figure 2 cells-13-00572-f002:**
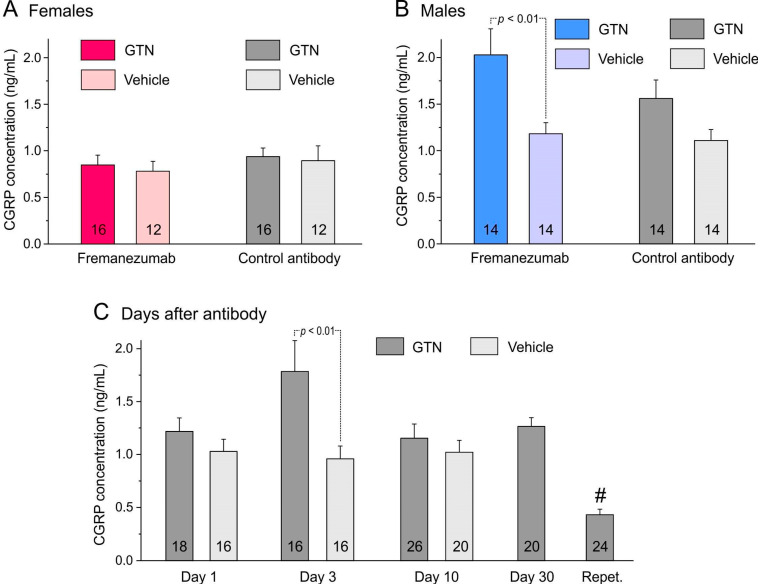
CGRP concentration (means ± SEM) of trigeminal ganglia separated by sexes (**A**,**B**), administration of fremanezumab or control mAb and treatment with GTN or vehicle. The difference between ganglia of GTN and vehicle-treated animals depends largely on males (**B**) with a significant difference between the fremanezumab groups (Tukey’s HSD post hoc test following factorial ANOVA, *p* < 0.01). (**C**): The difference between GTN and vehicle-treated groups (sex and mAb groups cumulated) is statistically significant on day 3 after fremanezumab/control mAb treatment (unequal N HSD post hoc test following factorial ANOVA, *p* < 0.01). The low CGRP concentration in the trigeminal ganglion samples of repetitively GTN-treated animals (Repet., #) is significantly different from all other GTN-treated groups (HSD post hoc test following factorial ANOVA, *p* < 0.05–0.0001). Numbers within bars mean number of trigeminal ganglia.

**Figure 3 cells-13-00572-f003:**
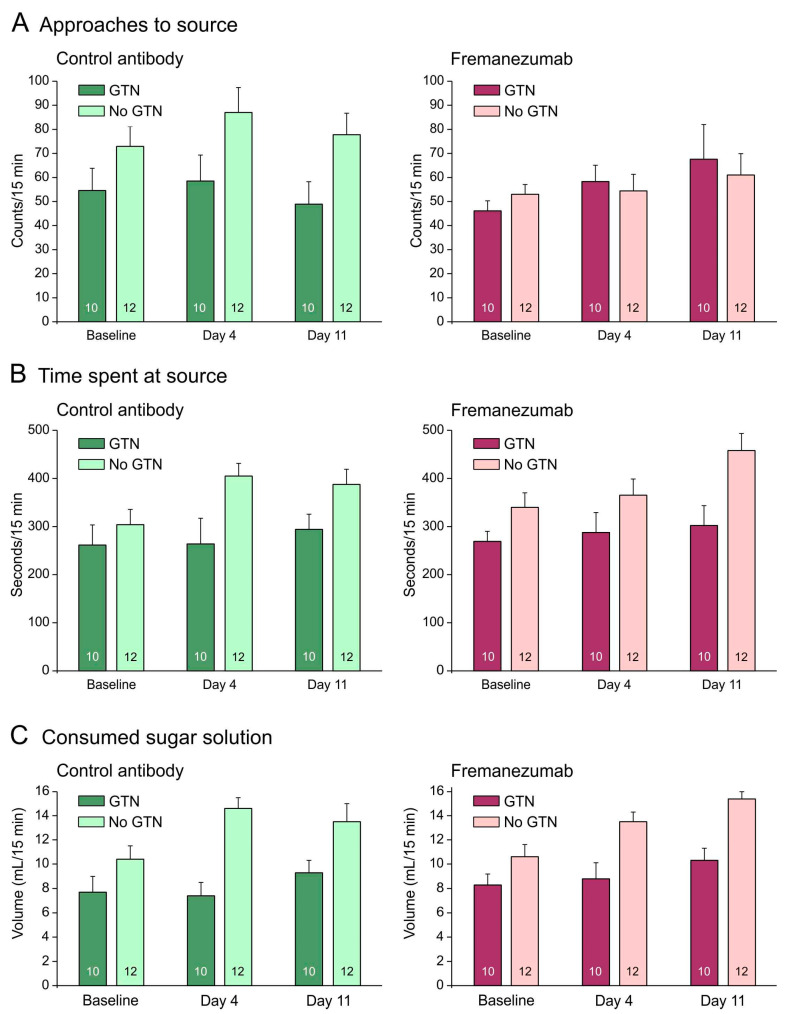
No barrier condition. Behavioral data (means ± SEM) of animals before (baseline) and days after administration of fremanezumab or control mAb and treatment or no treatment with GTN. Female and male data are not presented separately, because there was no sex difference. Animals treated with GTN tended to approach the attractive source less frequently (**A**), stayed less time at the source (**B**) and consumed less sugar solution (**C**) than animals not treated with GTN. There was no difference between fremanezumab and control mAb regarding any of these behavioral data. Numbers within bars mean number of animals.

**Figure 4 cells-13-00572-f004:**
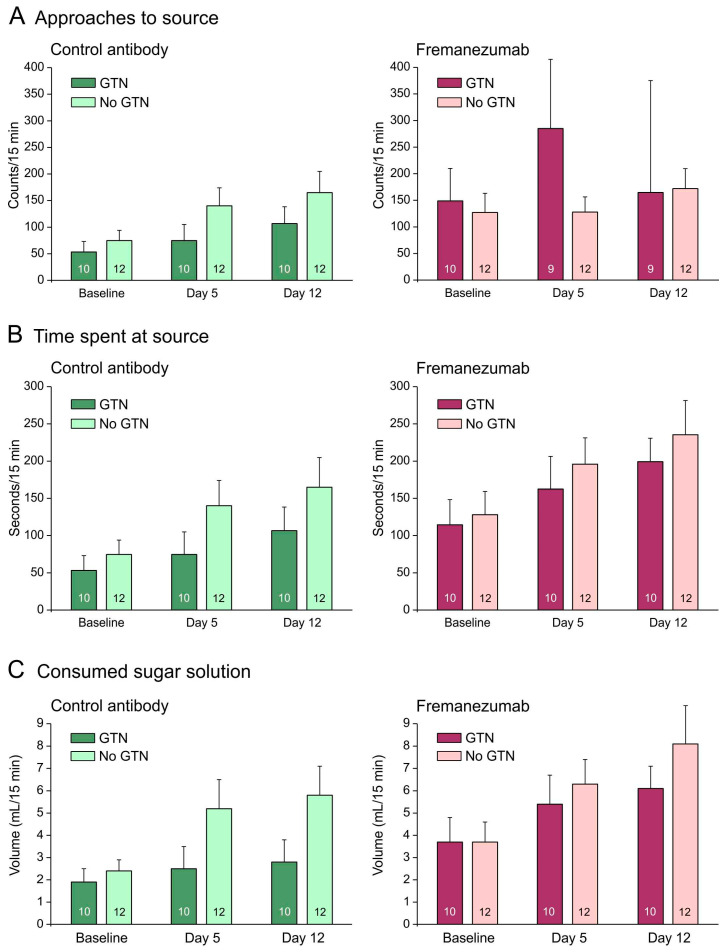
Mechanical barrier condition. Behavioral data of animals before (baseline) and days after administration of fremanezumab or control mAb and treatment or no treatment with GTN (means ± SEM). Female and male animal data are combined because there was no sex difference. Animals treated with GTN tended to approach the attractive source less frequently (**A**), to stay less time at the source (**B**) and to consume less solution (**C**) when they had received control mAb. Animals that received fremanezumab stayed longer at the source and consumed more volume compared to animals that received control mAb. Numbers within bars mean number of animals.

**Table 1 cells-13-00572-t001:** Trigeminal ganglion mass and CGRP concentration (means ± SEM) in ganglion samples comparing different groups of animals treated with fremanezumab or control mAb and GTN.

		Samples (n)	Ganglion Mass (mg)	Difference	CGRP Conc. (ng/mg)	Difference
mAb	Fremanezumab	52	14.7 ± 0.7	n.s.	1.11 ± 0.12	n.s.
	Control antibody	52	13.7 ± 0.8		1.10 ± 0.90	
Sex	Females	52	15.1 ± 0.9	*p* < 0.05	0.87 ± 0.56	*p* < 0.0001
	Males	52	13.4 ± 0.6		1.34 ± 0.13	
Days after mAb administration	Day 1	18	17.7 ± 1.1	*p* < 0.0001	1.22 ± 0.13	*p* < 0.0001
	Day 3	16	15.7 ± 1.6		1.71 ± 0.29	
	Day 10	26	14.7 ± 1.0		1.15 ± 0.13	
	Day 30	20	8.9 ± 0.4		1.25 ± 0.83	
	GTN repetitive	24	14.6 ± 1.0		0.44 ± 0.05	

GTN was applied on the indicated day. “GTN repetitive” animals were treated daily with GTN on days 4–6 and 11–13. Significant differences (*p*) result from factorial ANOVA comparing two or five variables, respectively; n.s., not significant.

**Table 2 cells-13-00572-t002:** Number of approaches (counts), time at the source and consumed volume (means ± SEM) in animals treated with fremanezumab or isotype control mAb and GTN (all groups, *n* = 10).

	Baseline		Day 4–6		Day 11–13	
No barrier	Fremanezumab	Control mAb	Fremanezumab	Control mAb	Fremanezumab	Control mAb
Counts (*n*)	46.1 ± 4.2	54.6 ± 9.2	58.3 ± 6.8	58.5 ± 10.4	67.6 ± 14.4	48.9 ± 4.6
Time (s)	269.0 ± 21.0	261.7 ± 41.6	287.8 ± 41.5	263.9 ± 53.3	302.1 ± 41.4	294.0 ± 29.2
Volume (mL)	8.3 ± 0.9	7.7 ± 1.3	8.8 ± 1.3	7.4 ± 1.1	10.3 ± 1.1	9.3 ± 1.0
Mech. barrier	Fremanezumab	Control mAb	Fremanezumab	Control mAb	Fremanezumab	Control mAb
Counts (*n*)	141.5 ± 55.3	44.8 ± 15.1	383.7 ± 152.5	113.9 ± 56.4	553.7 ± 210.6	121.5 ± 56.3
Time (s) *	114.5 ± 33.7	53.3 ± 19.8	162.6 ± 43.8	74.7 ± 30.2	199.2 ± 41.5	106.7 ± 32.6
Volume (mL) *	3.7 ± 1.1	1.9 ± 0.6	5.4 ± 1.3	2.5 ± 1.0	6.1 ± 1.4	2.8 ± 1.0
Therm. barrier	Fremanezumab	Control mAb	Fremanezumab	Control mAb	Fremanezumab	Control mAb
Counts (*n*)	64.8 ± 9.8	73.3 ± 12.9	66.7 ± 10.3	55.7 ± 11.3	82.9 ± 10.1	62.7 ± 11.5
Time (s)	228.9 ± 38.3	195.6 ± 34.0	230.2 ± 32.5	155.8 ± 41.3	220.3 ± 28.5	175.5 ± 37.8
Volume (mL)	6.8 ± 1.2	5.3 ± 1.0	6.8 ± 0.8	4.5 ± 1.0	6.6 ± 0.8	5.9 ± 1.1

Significant differences (repeated measures ANOVA, *p* < 0.05) between fremanezumab and control mAb (*) are seen under mechanical barrier conditions. Mech., mechanical; Therm., thermal.

## Data Availability

Data are contained within the article.

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
