# Peer review of "Glycerol Trinitrate Acts Downstream of Calcitonin Gene-Related Peptide in Trigeminal Nociception—Evidence from Rodent Experiments with Anti-CGRP Antibody Fremanezumab"

_cells, 2024, doi:10.3390/cells13070572_

Round 1

Reviewer 1 Report

Comments and Suggestions for Authors

Nicola Benedicter et al. found that glycerol trinitrate acts downstream of calcitonin gene-related peptide in trigeminal nociception by using the anti-CGRP antibody fremanezumab in rodent model.

The results were solid and of potential interests. I have some concern before it publication.

Major concern:

1.       For Table 2, label of the star “*” was not so clear.

2.       For Figure 3C, on Day 4, there was differences between GTN and No GTN groups. The authors need to show the statistical data.

3.       There was some significant sex differences found in Figure 2. Thus, the authors pooled data from male and female rats seems to be inappropriate.

Author Response

Nicola Benedicter et al. found that glycerol trinitrate acts downstream of calcitonin gene-related peptide in trigeminal nociception by using the anti-CGRP antibody fremanezumab in rodent model.

The results were solid and of potential interests. I have some concern before it publication.

Major concern:

  1. For Table 2, label of the star “*” was not so clear.

Labelling has been corrected.

  1. For Figure 3C, on Day 4, there was differences between GTN and No GTN groups. The authors need to show the statistical data.

Repeated measures ANOVA showed a general statistical difference between GTN and No GTN groups, which has been noted in the text (line 273-276). However, differentiating the groups according to the time points (i.e. baseline, day 4 and day 11), as it is displayed in Figure 3, data are not significantly different according to the Tukey post-hoc test. Instead of a graph summarizing all GTN versus No GTN data, we wanted to show the differentiated data and consequently added no significance star.  

  1. There was some significant sex differences found in Figure 2. Thus, the authors pooled data from male and female rats seems to be inappropriate.

Data from males and females were not pooled regarding fremanezumab versus control antibody (Figure 2A and B), because they were different between sexes according factorial ANOVA analysis, as noted (lines 215-216). Pooling was only applied for the differentiation according to the days after antibody administration (Figure 2C), where there was no significant sex difference.

Reviewer 2 Report

Comments and Suggestions for Authors

This study aimed to explore mechanisms of action and interaction between CGRP and nitric oxide (NO) as the two key plays in migraine, in facial hyperalgesia rats. One key finding is that elevation of trigeminal ganglion CGRP level post GTN treatment in fremaneuzumab male rats, but not, females. The Mechanistic study and explanation of the finding(s) are somewhat limited when addressing ‘glycerol trinitrate acts downstream of CGRP in trigeminal nociception’ (part of the title). Perhaps monitoring NO production/NOS activity in the tissue post the antibody injection would help to support the conclusion drawn. NO activates cGMP pathway while CGRP activate cAMP pathway. Discussion could be elaborated regarding possible independent mechanisms involved. 

Other comments are:

Line 93: What is the similarity of Fremanezumab target sequences between human and Wistar rat?

Line 84, how many rats in total were used?

Line 104: Experimental design. It would helpful to justify different doses (5 mg/kg, 2.5 mg/kg and 1.25 mg/kg GTN) applied for three consecutive days.

Line 109: Are the trigeminal ganglion of each rat, right, left or combined?

Line 125: Please indicate the amount of the TG homogenate used for ELISA

Line 127: Was the trigeminal ganglion left or right or combined tissue from each rat?

Line 144, Were the sequence of testing conducted in the mornings or afternoons? please indicate

Line 177:Please provide sample number along with mean +/-SE value

Line 184: CGRP concentration of left and right trigeminal ganglion from the same rat(s) were dependent samples. Were they averaged prior to statistical analysis between control/Fremanezumab groups (independent samples)?

Line 198-199. Table 1 legend is somewhat confusing. Were the animals (with sample number = 52) treated with GTN? Please provide more information,

Figure 2B. Were CGRP concentrations compared between Fremanezumab and control antibody groups in animals treated with GTN in males?

Line 325. Should it be Figure 2A, B, C? Please check the figure number stated is correct in the same paragraph.

Line313. Female rat TG had relatively lower CGRP levels than the male treated with GTN. Could the difference link with their sex-differences in hormone levels?

Author Response

This study aimed to explore mechanisms of action and interaction between CGRP and nitric oxide (NO) as the two key plays in migraine, in facial hyperalgesia rats. One key finding is that elevation of trigeminal ganglion CGRP level post GTN treatment in fremaneuzumab male rats, but not, females. The Mechanistic study and explanation of the finding(s) are somewhat limited when addressing ‘glycerol trinitrate acts downstream of CGRP in trigeminal nociception’ (part of the title). Perhaps monitoring NO production/NOS activity in the tissue post the antibody injection would help to support the conclusion drawn. NO activates cGMP pathway while CGRP activate cAMP pathway. Discussion could be elaborated regarding possible independent mechanisms involved. 

We thank the reviewer for his/her interesting comment. NO production/NOS activity is technically difficult to monitor and poorly reliable but NOS immunoreactivity would be a possible solution. However, we are not able to perform new experiments, since the maximal number of animals approved in our application is exhausted.

Regarding possible independent mechanisms involved: We have added some sentences to the end of the Discussion addressing the possibility that HNO (nitroxyl) rather than NO activating intracellular cGMP is involved in the cascade of nociceptive events: “The decisive NO species thereby may be nitroxyl (HNO), a redox sibling of NO formed in the presence of H2S, which is able to directly activate transient receptor potential receptor channels of the ankyrin type (TRPA1) in trigeminal afferents [48,49]. NO species have recently also been suggested to activate vanilloid-type transient potential receptor channels (TRPV1) through S-nitrosylation [50]. In any case, the classical intracellular NO-mediated mechanism increasing cyclic guanosine monophosphate [51] is not necessary to explain the activating and sensitizing effects of NO species in pain and migraine generation [52, 53]”.

Other comments are:

Line 93: What is the similarity of Fremanezumab target sequences between human and Wistar rat?

The epitope for fremanezumab is identical with respect to human alpha-CGRP, beta-CGRP and rat beta-CGRP. There is one similar amino acid change within the epitope for rat alpha-CGRP. This amino acid difference has a minor effect on overall binding affinity and fremanezumab has demonstrated good efficacy in preclinical rodent models. We have added to the Methods (2.2): “Fremanezumab binds to an epitope that is identical in human a-CGRP, b-CGRP and rat b-CGRP. There is one similar amino acid change within the epitope for rat a-CGRP, which has a minor effect on overall binding affinity.”

Line 84, how many rats in total were used?

In total, 52 animals were used. We have added the number to the Methods (paragraph 2.1).

Line 104: Experimental design. It would helpful to justify different doses (5 mg/kg, 2.5 mg/kg and 1.25 mg/kg GTN) applied for three consecutive days.

During the sequence of 3 days, measurements without barrier, then with mechanical barrier and finally with thermal barrier were performed after GTN injection. Assuming that repetitive GTN injections with the same dose would result in cumulating effects, for the following applications we reduced the doses to the half, which seemed to be a good compromise to achieve a stable effect. Nevertheless, there was probably some accumulation/sensitization regarding the decrease in ganglion CGRP concentration after this treatment. We have added to the Methods (paragraph 2.3): “With this design, a cumulative sensitizing effect of GTN should be avoided”.

Line 109: Are the trigeminal ganglion of each rat, right, left or combined?

From each animal left and right trigeminal ganglia were taken and independently analysed for their CGRP content. In the Methods, paragraph 2.3, the information is given: “From each animal, both trigeminal ganglia were harvested, resulting in 104 samples”. This procedure increases the variance, which is compensated by the higher sample number for statistical analysis. We have earlier demonstrated that there is no statistical difference regarding weight and CGRP content between left and right trigeminal ganglion in rats (ref. 29).

Line 125: Please indicate the amount of the TG homogenate used for ELISA

Each ganglion (masses see Table 1) was homogenized in 1 mL of acetic acid. This information was added to the Methods (paragraph 2.4).

Line 127: Was the trigeminal ganglion left or right or combined tissue from each rat?

Each ganglion was processed separately, see above.

Line 144, Were the sequence of testing conducted in the mornings or afternoons? please indicate

The sequence of testing was conducted always between 12 p.m. and 1 p.m. This information has been added to the Methods (paragraph 2.6).

Line 177:Please provide sample number along with mean +/-SE value

Sample numbers (n = 26 for fremanezumab and control antibody, respectively) have been provided.

Line 184: CGRP concentration of left and right trigeminal ganglion from the same rat(s) were dependent samples. Were they averaged prior to statistical analysis between control/Fremanezumab groups (independent samples)?

From each animal, left and right trigeminal ganglion was dissected and independently analysed for CGRP content. Although both ganglia can be regarded as dependent samples, the weight showed some variance due to the attached connective tissue. However, we have earlier demonstrated that there is no statistical difference regarding weight and CGRP content between left and right trigeminal ganglion (ref. 29). We could have averaged mass and CGRP content of both ganglia in each animal, thereby reducing both the sample size and the variance, which makes statistically no difference in the calculated data.

Line 198-199. Table 1 legend is somewhat confusing. Were the animals (with sample number = 52) treated with GTN? Please provide more information,

All animals, the data of which are listed in Table 1, were treated with GTN. This was added to the legend.

Figure 2B. Were CGRP concentrations compared between Fremanezumab and control antibody groups in animals treated with GTN in males?

Yes, CGRP concentrations were compared between these groups but there was no significant difference according to the post-hoc test.

Line 325. Should it be Figure 2A, B, C? Please check the figure number stated is correct in the same paragraph.

Yes, it should be Figure 2. We thank the reviewer for detecting this error.

Line313. Female rat TG had relatively lower CGRP levels than the male treated with GTN. Could the difference link with their sex-differences in hormone levels?

Yes, indeed we assume that sex hormones influence the expression or production of CGRP dependent on GTN. We have added to the Discussion (paragraph 4.1): “It is likely that sex hormones influence the expression or production of CGRP. It has long been reported that in rat dorsal root ganglia the number of CGRP immunoreactive neurons was lower in females and increased in ovariectomized but decreased in estradiol treated animals [36]. More recent experiments showed that 17β-estradiol decreased CGRP plasma levels and CGRP release from isolated trigeminal ganglia in male rats [37].”

Reviewer 3 Report

Comments and Suggestions for Authors

The study assesses the impact of a monoclonal antibody against CGRP, used as an anti-migraine treatment, on an animal model induced by the administration of glyceryl trinitrate (GTN). Male and female Wistar rats were treated with GTN at different times after administration of the anti-CGRP mAb fremanezumab or isotype control mAb. Mechanical and thermal allodynia (expressed as cumulative time spent at the source to consume the sugar solution, time and volume consumed) were assessed in one group of animals using the orofacial stimulation test. CGRP levels in the trigeminal ganglia (TGs) were measured. Administration of a single dose of GTN increased CGRP levels in the TGs 4 hours later. By contrast, administration of multiple doses of GTN over several days decreased CGRP levels, probably related to its release. CGRP levels in the TGs of GTN-treated rats were lower in females than in males. GTN animals also tended to spend less time at the source and to consume less sugar solution than the control group. Data from male and female animals were combined, as there was no difference between the sexes. GTN and monoclonal vehicle-treated animals exhibited reduced attraction towards the sugar source. By contrast, GTN-treated animals injected with fremanezumab remained at the food source longer and consumed more food than the control group. There was no significant difference in CGRP levels in the TGs after treatment with the monoclonal antibody. The results suggest that nitric oxide may act downstream of CGRP signaling in the pathophysiology of migraine.

The study investigates the relationship between nitric oxide and CGRP in the pathogenesis of migraine and associated symptoms. However, there are issues with the manuscript that should be addressed before publication.

1)    The authors should clearly state the dosage of GTN used in their migraine model. In the Materials and Methods section, the dosage of GTN is 5 mg/kg, which differs from the dose reported in the abstract.

2)    GTN is insoluble in saline. Why is the solubility of 1mg in 1ml of saline reported?

3)    The experimental design is not intuitive, and the pharmacological doses need to be supported by references. Please to explain the reason of the 3 consecutive GTN injection with lowering doses. Why the animals were anesthetized (line 102)? 

4)    A sample size calculation is not indicated. Please to provide it together with the primary outcome.

5)    The study shows that after treatment with the monoclonal antibody, despite the reduction in trigeminal allodynia, no change in CGRP levels was reported in the TGs of the rats. Perhaps it is worth making a hypothesis. A recent clinical study (DOI:10.3390/pharmaceutics15010293) investigated circulating CGRP levels after treatment with the monoclonal antibody. The authors showed that patients treated with CGRP(-R) mAbs for at least eight months had similar circulating plasma CGRP concentrations to those without prior treatment. That's probably worth mentioning.

Minor

Statistical significance must be reported in the graphs when it is present. Some bibliographic entries require organization by journal.

Author Response

The study assesses the impact of a monoclonal antibody against CGRP, used as an anti-migraine treatment, on an animal model induced by the administration of glyceryl trinitrate (GTN). Male and female Wistar rats were treated with GTN at different times after administration of the anti-CGRP mAb fremanezumab or isotype control mAb. Mechanical and thermal allodynia (expressed as cumulative time spent at the source to consume the sugar solution, time and volume consumed) were assessed in one group of animals using the orofacial stimulation test. CGRP levels in the trigeminal ganglia (TGs) were measured. Administration of a single dose of GTN increased CGRP levels in the TGs 4 hours later. By contrast, administration of multiple doses of GTN over several days decreased CGRP levels, probably related to its release. CGRP levels in the TGs of GTN-treated rats were lower in females than in males. GTN animals also tended to spend less time at the source and to consume less sugar solution than the control group. Data from male and female animals were combined, as there was no difference between the sexes. GTN and monoclonal vehicle-treated animals exhibited reduced attraction towards the sugar source. By contrast, GTN-treated animals injected with fremanezumab remained at the food source longer and consumed more food than the control group. There was no significant difference in CGRP levels in the TGs after treatment with the monoclonal antibody. The results suggest that nitric oxide may act downstream of CGRP signaling in the pathophysiology of migraine.

The study investigates the relationship between nitric oxide and CGRP in the pathogenesis of migraine and associated symptoms. However, there are issues with the manuscript that should be addressed before publication.

  • The authors should clearly state the dosage of GTN used in their migraine model. In the Materials and Methods section, the dosage of GTN is 5 mg/kg, which differs from the dose reported in the abstract.

The dosage of GTN was 5 mg/kg body weight. We have corrected this error in the Abstract and thank the reviewer for detecting the error.

  • GTN is insoluble in saline. Why is the solubility of 1mg in 1ml of saline reported?

We thank the reviewer for this question and correct this information. We have used commercially available GTN (Nitrolingual, containing 1 mg/mL GTN, Pohl-Boskamp, Hohenlockstedt, Germany) and synthetic interstitial fluid (SIF) as vehicle. We have changed this information in the Methods (paragraph 2.3) and added the reference [27] where the composition of SIF is reported. SIF is a balanced neutral buffered solution composed of several electrolytes as found in extracellular solutions. We avoided using saline as vehicle, because in former experiments with electrophysiological recordings from trigeminal neurons, i.p. injection of saline of more than 1 mL caused sometimes an excitation, which does not happen with SIF.

  • The experimental design is not intuitive, and the pharmacological doses need to be supported by references. Please to explain the reason of the 3 consecutive GTN injection with lowering doses. Why the animals were anesthetized (line 102)?

To clarify this point, we have added to the Discussion (paragraph 4.1): “In rodent models of migraine or facial pain, mostly 10 mg/kg GTN have been applied as a single dose [31, 32] or repetitively 5 mg/kg every second day [33]. Repetitive GTN injection in rats has been reported to induce spinal hyperalgesia and orofacial allodynia along with an increase in CGRP expression [30].”

In our behavioural measurements during the sequence of 3 days, i.e. measurements without barrier, then with mechanical barrier and finally with thermal barrier, we wanted to avoid as far as possible the generation of a persisting hyperalgesia, since our first aim was to see a possible effect of the anti-CGRP antibody. Assuming that repetitive GTN injections with the same dose of 5 mg/kg would result in cumulating effects influencing both behaviour and CGRP expression, we reduced the doses to the half for the second and again to the half for the third day, which seemed to be a good compromise to achieve a stable effect for measuring an influence of fremanezumab. Nevertheless, there was probably some accumulation/sensitization regarding the decrease in ganglion CGRP concentration after this treatment. We have added to the Methods (paragraph 2.3): “With this design, a cumulative sensitizing effect of GTN causing chronic hyperalgesia [30] should be avoided”.

The short anaesthesia for the GTN injection was performed according to the protocol approved by the ethics committee mentioned in the first paragraph of the Methods.

  • A sample size calculation is not indicated. Please to provide it together with the primary outcome.

We have added to the Methods (paragraph 2.7): "Sample size calculation was based on the biometric planning for previous measurements [27, 28] in accordance with the ethics approval mentioned above."

  • The study shows that after treatment with the monoclonal antibody, despite the reduction in trigeminal allodynia, no change in CGRP levels was reported in the TGs of the rats. Perhaps it is worth making a hypothesis. A recent clinical study (DOI:10.3390/pharmaceutics15010293) investigated circulating CGRP levels after treatment with the monoclonal antibody. The authors showed that patients treated with CGRP(-R) mAbs for at least eight months had similar circulating plasma CGRP concentrations to those without prior treatment. That's probably worth mentioning.

CGRP plasma concentrations below 100 pg/mL are far too low to effectively activate CGRP receptors, probably they represent the spontaneous ongoing release from the innervated tissues. If there is no impact of CGRP mAbs on the content of CGRP in trigeminal ganglia (and probably also in other tissues), one would not expect that the plasma CGRP levels change significantly. Although we have made no plasma CGRP measurements in the present study, we appreciate the reviewer’s question and have added a small additional paragraph to the discussion (4.3. No impact of fremanezumab on the CGRP concentration of trigeminal ganglia): “We have seen no significant difference in CGRP concentration between ganglia of rats treated with fremanezumab versus control antibody, which is consistent with our previous findings, although it contrasts with a decrease in CGRP immunoreactive trigeminal ganglion neurons [29]. The lacking change in CGRP concentration is reminiscent to a recent clinical study, in which plasma CGRP levels in patients were determined during the treatment with fremanezumab and four months after discontinuation of the treatment [38]. CGRP plasma concentrations did not differ between treatment and post-treatment and were on the same level as in healthy controls. Thus it seems likely that the main therapeutic effect of monoclonal antibodies directed against CGRP depends rather on a decrease in CGRP release during strong stimulation of trigeminal afferents, as we have demonstrated earlier [27]. On the other hand, long-term effects of anti-CGRP monoclonal antibodies may be based on a decrease in CGRP receptors, which is suggested by a significant reduction in trigeminal ganglion neurons immunoreactive to the CGRP receptor components RAMP1 and CLR [29].”                    

Minor

Statistical significance must be reported in the graphs when it is present. Some bibliographic entries require organization by journal.

All statistical significances have been reported in the graphs by * or #. However, in some cases there was statistical significance of whole groups (e.g. between GTN and vehicle treatment), which was no longer the case when the groups were differentiated (e.g. in male and female groups, only different for males) and therefore not indicated in the specified graphs.

Round 2

Reviewer 2 Report

Comments and Suggestions for Authors

The authors addressed all my questions reasonably.

Reviewer 3 Report

Comments and Suggestions for Authors

The authors responded to all comments